# Assessment of Metabolic Syndrome and Kidney and Heart Function in Childhood Cancer Survivors

**DOI:** 10.3390/children10061073

**Published:** 2023-06-18

**Authors:** Aleksandra Janecka, Joanna Stefanowicz, Anna Owczarzak, Marek Tomaszewski, Tomasz Batko, Ninela Irga-Jaworska

**Affiliations:** 1Department of Paediatrics, Haematology and Oncology, Faculty of Medicine, Medical University of Gdansk, Debinki 7, 80-211 Gdańsk, Poland; aleksandra.janecka@gumed.edu.pl (A.J.); ninela.irga-jaworska@gumed.edu.pl (N.I.-J.); 2Department of Paediatrics, Haematology and Oncology, University Clinical Centre, Debinki 7, 80-952 Gdansk, Poland; mtomaszewski@uck.gda.pl (M.T.); tbatko@uck.gda.pl (T.B.); 3Faculty of Health Sciences with Institute of Maritime and Tropical Medicine, Medical University of Gdansk, 80-211 Gdansk, Poland; 4Department of Clinical Nutrition, Faculty of Health Sciences with Institute of Maritime and Tropical Medicine, Medical University of Gdansk, Debinki 7, 80-211 Gdansk, Poland; annaow@gumed.edu.pl

**Keywords:** childhood cancer survivors, long-term side effects, metabolic disorders, hypertension, salusin β

## Abstract

Background: The survivors of childhood cancer suffer from a number of long-term side effects. These include atherosclerosis and cardiovascular diseases (CVDs) that develop earlier in adulthood than in the rest of the population. The aim of this study was to identify prognostic factors of developing atherosclerosis before the development of symptomatic CVD. Methods: A total of 141 children that were 7–18 years old were examined; there were 116 survivors of childhood malignancies (hematopoietic and lymphoproliferative malignancies—52; neuroblastoma—22; Wilms tumor—24; other solid tumors—18) and 25 healthy controls. Anthropometric measurements, blood pressure measurements, ultrasonography of the abdomen, echocardiography, and laboratory tests were performed. Results: There were no significant differences in gender distribution, time from the end of the treatment, weight, BMI, prevalence of central obesity, blood pressure and resistive index of the renal arteries, lipid profile, or glucose and fibrinogen levels. Patients with solid tumors had a significantly lower height and worse renal function. Patients with hematological malignancies significantly presented the lowest shortening fraction of the left ventricle. The salusin β levels were significantly higher in the control group than among the patients. Conclusions: The type and severity of side effects are closely related to the type of neoplasm and the treatment that has been undergone. Careful observation and regular follow-up are necessary.

## 1. Introduction

Over recent decades, it has been possible to observe significant developments in medicine as well as in oncological treatment for children. Today, we pay attention not only to curing cancer but also to the quality of life after treatment. Consequently, late side effects are an area of interest for researchers. Late side effects include pulmonary, cardiovascular, gastrointestinal, genitourinary, musculoskeletal, and endocrinological disorders, neurological complaints, nutritional disorders, short stature, infertility, immune deficiency, and a second neoplasm. The majority (60–90%) of survivors report at least one complaint. The type of symptom mostly depends on the type and localization of the neoplasm, the treatment (chemotherapy, radiotherapy, or surgery), and the treatment era. Progress in medicine results in better curability and fewer severe late side effects. The time from the end of the treatment and the age at the moment of diagnosis also seem to have an impact on the reported disorders [1,2,3,4].

Nevertheless, circulatory problems are a fairly common and severe complication. They occur in 20–30% of patients [3,5].

They involve hypertension, heart failure, coronary artery disease, and cerebrovascular disorders. Cytostatics—especially anthracyclines—and radiotherapy are cardiotoxic agents. Moreover, oncological treatment seems to accelerate the development of atherosclerosis. Consequently, childhood cancer survivors (CCSs) have a risk of cardiac death that is seven times higher than that in the general population [6,7,8].

At the same time, cardiovascular diseases (CVDs) are responsible for most deaths in adults. The risk factors of CVDs are well known. On this basis, the term “metabolic syndrome” was established. The components of metabolic syndrome are obesity, hypertension, hyperglycemia, and lipid disorders [9].

Obesity in CCSs that develops in childhood, adolescence, or young adulthood is strongly associated with the subsequent development of health problems later in life, such as adult-onset diabetes mellitus, hypertension, dyslipidemia, cardiovascular disease, and osteoarthritis [10]. Obesity is a well-recognized consequence of childhood cancer treatment. The prevalence of obesity in childhood ALL survivors ranges from 11% to 40%. Compared with age-, sex-, and race-matched controls, adult survivors of childhood ALL have a 1.5-fold excess risk of obesity in comparison with the general population [11].

The prognostic factors were researched. Among them, salusin β seems to be a promising agent. Most studies revealed increased serum levels of salusin β in patients with diabetes, hyperlipidemia, hypertension, and obesity [12,13,14].

The aim of our study was to identify the prognostic factors of developing atherosclerosis before the development of symptomatic CVDs.

## 2. Patients and Methods

### 2.1. Patients

This study enrolled 141 children ages 7 to 18 years, including 116 survivors of childhood malignancies and 25 healthy controls. The patients were treated in the Department of Paediatrics, Haematology, and Oncology at the Medical University of Gdansk from 1999 to 2017. The examinations in the study were conducted at least 5 years after the end of oncological treatment during routine controls.

Patients who had undergone any kind of transplantation were excluded.

The study consisted of an assessment of patient history and physical examinations, including anthropometric measurements, triple blood pressure (BP) measurements, ultrasonography of the abdomen, echocardiography, and the collection of blood and urine samples.

The survivors were divided into 4 groups according to the diagnosis that determined the treatment: hematopoietic and lymphoproliferative malignancies (*n* = 52), neuroblastomas (*n* = 22), Wilms tumors (*n* = 24), and other solid tumors (*n* = 18).

The group with hematopoietic and lymphoproliferative malignancies comprised patients with acute lymphoblastic leukemia (ALL) (*n* = 37), acute myeloid leukemia (AML) (*n* = 1), Hodgkin lymphoma (HL) (*n* = 3), non-Hodgkin lymphoma (NHLs) (*n* = 6), and Langerhans cell histiocytosis (LCH) (*n* = 5). All of them underwent standard chemotherapy: 8 patients with ALL and 1 with AML received radiotherapy for the central nervous system; 1 patient with HL was treated with radiotherapy to the mediastinum; and 1 patient with NHL underwent radiotherapy to the abdomen.

In the group with Wilms tumors (WTs), 5 patients were treated with radiotherapy (4—abdomen, 1—lungs). In the group with neuroblastoma, 1 patient was treated with radiotherapy. In two patients with WTs, bilateral tumors were diagnosed; they underwent patrial nephrectomy. The rest of the children with WTs were treated with total nephrectomy.

The group with solid tumors consisted of patients with rhabdomyosarcoma (*n* = 7), hepatoblastoma (*n* = 4), yolk sac tumors (YSTs) (*n* = 3), other germ cell tumors (GCTs) (n=1), retinoblastoma (*n* = 1), alveolar soft-part sarcoma (*n* = 1), and medulloblastoma (*n* = 1). Six patients underwent radiotherapy: five with rhabdomyosarcoma and one with medulloblastoma.

### 2.2. Methods

The anthropometric measurements—such as measurements of the height, weight, and waist circumference of the patients—were performed using standard techniques (WE150, Mensor, Warsaw, Poland).

The laboratory blood testing included evaluation of the blood count, serum creatinine, cystatin C, glucose, lipid profiles, fibrinogen, uric acid, and salusin β levels after an overnight fast.

Urinalysis and albumin and creatinine excretion were performed using urine sampling. Furthermore, the albumin-to-creatinine ratio (ACR) was evaluated.

The serum creatinine was detected using an enzymatic method (Alinityc Creatinine Reagent Kit, Abbott). The serum cystatin C levels were analyzed using immunonephelometry (N Latex Cystatin C, Siemens). The estimated glomerular filtration rate (eGFR) was calculated based on the serum creatinine and cystatin C levels.

The eGFR was indirectly measured by using the original Schwartz-, creatinine-, and BUN-based equations and Filler formulas.

The Schwartz formula was evaluated as follows: GFR in mL/min/1.73 sq m = k × height in cm/serum creatinine concentration in mg/dL, where the constant k = 0.413 originated from the values for children published in the literature. [15]. The creatinine- and BUN-based eGFR was calculated according to the following equation: 40.7(height/SCr)0.64(30/BUN)0.202 [16].

Additionally, the GFR was calculated based on cystatin C concentration according to the Filler formula: logGFR = 1.962 + [1.123 × log(1/cystatin C)] [17].

The plasma lipid profile was defined by electrophoresis (Hydragel 15 Lipo + Lp(a) Sebia). The concentration of salusin β (pg/mL) was assayed by using an immunoenzymatic technique with an ELISA kit for salusin β (produced by Cloud-Clone Corp., Houston, TX, USA; Wuhan, China, 2018).

Metabolic syndrome and central obesity were defined according to the criteria of the International Diabetes Federation [18].

Blood pressure (BP) was measured three times in every child in the study by using an oscillometric method with a standard clinical sphygmomanometer (professional blood pressure monitor: HBP-1100-E, OMRON Healthcare Co., Ltd., Kyoto, Japan, 2014), according to the guidelines and recommendations of the Polish Pediatric Nephrology Society. The mean values of the systolic and diastolic pressures were determined. The results were then compared to the reference values provided by the OLAF project, which were matched according to sex, age, and height [19,20].

Ultrasonography of the abdomen was performed by well-trained and experienced sonographers according to a standardized protocol (Toshiba Aplio 500, Philips Epiq 7G, Fujifilm Arietta 750). During the examination, the kidneys were assessed. Moreover, a spectral Doppler assessment was conducted in the arcuate arteries of the kidneys to evaluate the renal arterial resistive index (RI). Transthoracic echocardiography was performed with a single echosonographer and cardiologist according to the recommendations of the American Society of Echocardiography (Canon Xario 200G) [21].

### 2.3. Statistical Methods

The mean, median, standard deviation, range (minimum–maximum), and lower and upper quartiles (25 Q and 75 Q) were calculated for each parameter. The statistical significance between means for different groups was calculated by using a one-way analysis of variance (ANOVA) or by using the nonparametric Mann–Whitney U or Kruskal–Wallis test when the T-variances in the groups were not homogeneous (the homogeneity of the variance was determined by using Bartlett’s test).

Statistical significance between frequencies was calculated by using the chi-square test (χ^2^df) with Yate’s correction and with the corresponding degree of freedom (df; df = (m − 1) × (n − 1), where m indicates the number of rows, and n indicates the number of columns).

The relationships between the two parameters were assessed using correlation analysis, and the Spearman correlation coefficients were calculated.

For the reference population, we used the results of the OLAF study, which was performed on children from the Polish population who were aged 7 to 18 years.

A *p*-value of less than 0.05 was required to reject the null hypothesis. The statistical analysis was performed using the EPIINFO Ver. 7.1.1.14 software package.

### 2.4. Ethics Committee

This study was accepted by the Independent Bioethics Committee for Scientific Research at the Medical University of Gdansk (NKBBN/359/2015, NKBBN/359-58/2018, NKBBN/359-9/2019, NKBBN/359-721/2021). Written informed consent was received from the legal guardians of the children. The procedures followed were in accordance with the Declaration of Helsinki of 1975, as revised in 2000.

## 3. Results

The evaluated groups varied in age. The group with neuroblastoma consisted of the youngest patients. The healthy volunteers were significantly younger than the patients with hematopoietic and lymphoproliferative malignancies and older than the patients with neuroblastoma.

There were no significant differences in gender distribution, time from the end of the oncological treatment, weight, BMI, or central obesity, which were evaluated in percentiles.

There was a significant difference in height. Patients who had undergone treatment for solid tumors had the shortest height in comparison with all other groups, and this difference was statistically significant. Patients who had undergone treatment for neuroblastoma were the tallest and were significantly taller than patients with hematological malignancies, Wilms tumors, and solid tumors; see Table 1.

Most of the patients reported the occurrence of cardiovascular disorders in their family members. In the group with solid tumors, CVDs were the rarest in their family histories.

Side effects during oncological treatments, such as kidney injuries and hypertension, were the most common in the WT patients; see Table 2.

There were no significant differences in either systolic or diastolic blood pressure in any of the examined groups; see Table 3.

Elevated blood pressure exceeding the 95th percentile in all measurements was revealed in 19% of the patients with hematological malignancies and in 17% of the patients with solid tumors; see Table 3.

The serum salusin β levels were significantly higher in the control group. The levels of uric acid were the highest in patients with hematopoietic malignancies and WTs. There were no statistically significant differences in the lipid profile or glucose and fibrinogen levels, as seen in Table 4. However, the Spearman correlation revealed a significant positive relationship between salusin β and low-density lipoprotein (LDL), as well as between cystatin C and ACR. There were no correlations between salusin β and age, time from the end of the treatment, weight, height, waist circumference, BMI, blood pressure, other lipid fractions, glucose, or eGFR.

Differences were also revealed in the kidney parameters in both the laboratory tests and the ultrasound examinations. Patients with solid tumors presented the worst kidney function, which was confirmed by their levels of creatinine, cystatin C, BUN, eGFR, and excretion of albumin in urine; see Table 5.

The ultrasound examinations revealed enlarged kidneys in most of the WT patients, and this occurred statistically more often than in other groups.

There were no significant differences in the RI of the renal arteries between the evaluated groups; see Table 6.

In echocardiography, the patients and the control group had similar ejection fractions (EFs), but the shortening fraction (SF) was significantly worse in the group with hematological malignancies than in the control group, the WT group, and the group with other solid tumors.

Metabolic syndrome was diagnosed in two patients (with a germ cell tumor and hepatoblastoma). Sixteen patients and two volunteers met two of the criteria for metabolic syndrome; see Table 7.

## 4. Discussion

Among the survivors of childhood malignancies, a number of late side effects are observed. These survivors have an increased risk of earlier onset of age-related chronic diseases, such as obesity, hypertension, dyslipidemia, diabetes mellitus, metabolic syndrome, cardiovascular diseases, and chronic kidney disease [2,22].

In our study, we concentrated on the components of metabolic syndrome and kidney parameters that are closely related to hypertension, and we determined one of the components of metabolic syndrome. Moreover, we measured the serum levels of salusin β as a predictor of the development of atherosclerosis.

According to the rules, we diagnosed metabolic syndrome in two patients with solid tumors. Although the number of patients evaluated was quite fair, it was not large enough to define any tendencies. We examined the biochemical parameters of metabolic syndrome—lipids, glucose, uric acid, and fibrinogen. We only detected higher concentrations of uric acid in survivors of hematological malignancies.

The patients differed in terms of waist circumference—the greatest difference was in the survivors of hematological malignancies. We did not reveal any significant differences in weight, BMI, or prevalence of central obesity between the groups of patients and the control group. Among the anthropometric measurements, only height was a significantly different feature among the patients. Children with solid tumors were shorter than the rest of the patients and the healthy controls. Patients with NBL were taller than the rest of the patients.

Why? What influenced the results?

The keys to answering this question are the type of cancer, risk group, and treatment that the children received, which depended on their diagnosis and risk group. The group of neuroblastoma survivors (NBLs) mostly consisted of patients with low and intermediate risk. Patients with high-risk NBLs are treated very aggressively with induction chemotherapy, surgery, mega-dose chemotherapy, autologous bone marrow transplantation (auto-BMT), radiotherapy, and immunotherapy. Our patients were mostly treated for a shorter time, and chemotherapy and surgery were standard treatments. The group with solid tumors other than NBLs and WTs included patients with RMS, HBLs, and GCTs. These patients received ifosfamide and cisplatin, and their chemotherapy was very aggressive and long.

Severe treatments for solid tumors also result in other side effects. Patients with solid tumors—apart from patients with Wilms tumors—presented the worst renal function. The survivors of solid tumors received nephrotoxic treatment, while in the survivors of Wilms tumors, this was a consequence of nephrectomy. These facts were confirmed in other studies [23,24,25].

The group with hematological malignancies stood out in terms of their age at the moment of diagnosis and their shortening fraction. This group was the oldest at the beginning of the treatment. The worse SF seemed to be a consequence of the anthracycline therapy that they received [26,27,28,29]. The EF remained at the same level in all evaluated groups.

Therefore, we divided the survivors into four groups according to their diagnoses: hematological malignancies, NBLs, WTs, and other solid tumors. The studied groups differed in age; the neuroblastoma survivors were younger than the others. This was compatible with epidemiological data—neuroblastoma is the most common tumor in infants and neonates.

The groups of patients that were compared were similar in terms of the time from the end of the treatment. However, patients with NBLs were the youngest at the moment of diagnosis, and patients with hematological malignancies were the oldest. The collected data did not reflect a dependence of the age distribution on epidemiology.

Previous studies suggested that patients who are treated in early childhood are more vulnerable to side effects than teenagers and young adults [4]. We cannot confirm this statement because we collected data from patients up to 11 years from the moment of diagnosis.

A study that Krawczuk-Rybak et al. performed on 1761 Polish CCSs reported short stature/obesity in over 20% of the survivors, and in general, disorders of the circulatory system were present in over 30% of the patients 5 years or more after the treatment; however, the authors did not provide information about the presented complaints in relation to the type of neoplasm [3].

However, Lipschultz et al. compared the conditions of 164 survivors in the Children’s Oncology Group with a matched noncancer population in the USA. The report revealed that obesity was significantly more common in the control group. Moreover, at least two criteria for metabolic syndrome were present significantly more often in the noncancer population, although the survivors had higher blood pressure and an increased frequency of prehypertension/hypertension. The prevalence of diabetes and dyslipidemia was similar in the survivors and the matched population. However, the survivors had lower predicted cardiovascular risk because their lifestyles were more favorable than those of the control group; the risk of future symptomatic cardiovascular disease predicted in childhood-cancer-survivor-specific risk models was moderate or higher [7].

Kooijmans et al. evaluated the renal function in 1024 Dutch adult CCSs at least 5 years after their treatment, without a division according to the type of neoplasm, and they performed a comparison with healthy controls. They revealed an increased risk of kidney dysfunction in CCSs, particularly at ages over 40 [30]. However, the prevalence of hypertension was similar between the two examined groups [31].

We assessed BP and ultrasound kidney parameters, and we confirmed that patients with Wilms tumors had large kidneys (expressed as the percentile depending on age and height), and this was connected with compensatory hypertrophy. However, we did not observe a difference in the renal arterial resistive index (RI).

A study by Merzenich et al. examined 4505 survivors of childhood cancers who were registered in the German Childhood Cancer Registry 5 years after their initial diagnosis in comparison with a healthy population; this confirmed that CCSs had an increased risk of death and, specifically, cardiac death [32]. This study followed earlier cohort reports that were performed in the United States/Canada, the UK, Switzerland, and the Nordic countries [33,34,35].

Recently performed research on CCSs provided quite consistent data. Such studies leaned towards the hypothesis of a higher risk of atherosclerosis and cardiac death in CCSs than in the general population. In recent decades, oncological treatments have been significantly modified, which has brought, at the same time, greater effectiveness and fewer side effects. In addition, medical care after treatment and the condition of the general population also influence the condition of CCSs. In developed countries, where medical care is well organized, diseases of civilization are leading health problems and reflect the long-term side effects of oncological treatment.

Our study is probably the first to evaluate the levels of serum salusin β in children after oncological treatment. The available literature has provided information about salusin β in children with hypertension, obesity, lipid disorders, and Down syndrome [36,37,38,39]. The publications concerning adult patients are more numerous and concentrate on symptomatic atherosclerosis or conditions directly leading to atherosclerosis, such as diabetes, hypertension, dyslipidemia, and obesity [40,41,42,43]. The type of oncological treatment also seems to be a factor of higher risk of atherosclerosis, but this was not reflected in the levels of salusin β.

Such a result was quite surprising for us, but it suggests that current knowledge about salusin β is still deficient.

This study has some limitations. We used quite a large population, but the groups were very diverse. Still, the group that was evaluated was too small to draw conclusions about population risk. The collected data consisted of information on patients from up to 11 years from the moment of diagnosis, and they do not reflect epidemiology. Moreover, the results are partially based on information obtained from patients’ parents (family history, side effects observed during the treatment).

## 5. Conclusions

Patients treated because of childhood cancers are at risk for long-term side effects. The type and severity of these side effects are closely related to the type of neoplasm and the treatment. Careful observation and regular follow-up in adulthood are necessary.

There is a need to conduct further studies to confirm the role of salusin β in the pathogenesis of atherosclerosis.

## Figures and Tables

**Table 1 children-10-01073-t001:** The characteristics of survivors and the control group; the median results (25Q ÷ 75Q) are provided.

	Control Group (*n* = 25)	Hematopoietic and Lymphoproliferative Malignancies (*n* = 52)	NBL ^2^ (*n* = 22)	WT ^3^ (*n* = 24)	Other Solid Tumors (*n* = 18)	*p*
Sex (M/F)	14/11	34/18	15/7	10/14	12/6	0.276
Age at the moment of examination (years)	11 (9.0 ÷ 14.0)	14.0 (12.0 ÷ 16.0)	8.5 (7.0 ÷ 12.0)	11.5 (10.0 ÷ 15.0)	12.5 (10.0 ÷ 15.0)	0.000 *
Age at the moment of diagnosis (years)		3.0 (2.0 ÷ 6.0)	0 (0 ÷ 1.0)	2.0 (1.0 ÷ 4.0)	2.0 (1.0 ÷ 5.0)	0.000 *
Time from the end of treatment (years)		7.0 (6.00 ÷ 9.0)	7.0 (6.0 ÷ 10.0)	7.5 (6.0 ÷ 10.0)	8.5 (6.0 ÷ 11.0)	0.853
Weight (pc ^1^)	63.0 (40.0 ÷ 89.0)	64.5 (50.5 ÷ 88.5)	68.0 (45.0 ÷ 88.0)	66.0 (35.0 ÷ 79.5)	36.0 (8.0 ÷ 77.0)	0.216
Height (pc ^1^)	74.0 (47.0 ÷ 90.0)	58.5 (32.0 ÷ 87.0)	90.0 (54.0 ÷ 96.0)	54.0 (36.5 ÷ 80.0)	26.0 (11.0 ÷ 63.0)	0.004 *
BMI (pc ^1^)	56.0 (26.0 ÷ 85.0)	69.5 (34.5 ÷ 87.5)	47.0 (35.0 ÷ 82.0)	60.5 (25.0 ÷ 85.0)	45.5 (22.0 ÷ 69.0)	0.482
Waist circumference (cm)	62.0 (57.0 ÷ 72.0)	73.0 (67.0 ÷ 81.0)	65.0 (57.5 ÷ 70.5)	65.0 (63.0 ÷ 71.5)	67.3 (60.0 ÷ 86.0)	0.002 *
Waist circumference <90 pc ^1^/>90 pc ^1^	91.3%/8.7%	73.47%/26.53%	80%/20%	80%/20%	61.54%/38.46%	0.288

* Statistically significant differences. ^1^ pc—percentile. ^2^ NBL—neuroblastoma. ^3^ WT—Wilms tumor.

**Table 2 children-10-01073-t002:** Risk factors for the components of metabolic syndrome in the survivors’ histories.

	Hematopoietic and Lymphoproliferative Malignancies	NBL ^1^	WT ^2^	Other Solid Tumors	*p*
Positive family history	75.51%	83.33%	100%	57.14%	0.083
Steroid therapy	98%	0%	0%	6.67%	0.000 *
Hypertension	7.84%	6.25%	58.85%	0%	0.000 *
Episode of acute kidney injury	1.96%	11.11%	25%	0%	0.403

* Statistically significant differences. ^1^ NBL—neuroblastoma. ^2^ WT—Wilms tumor.

**Table 3 children-10-01073-t003:** Blood pressure in the control group and in the survivor groups; the median results (25Q ÷ 75Q) are provided.

	Control Group	Hematopoietic and Lymphoproliferative Malignancies	NBL ^2^	WT ^3^	Other Solid Tumors	*p*
SBP * pc ^1^	54.0 (38.0 ÷ 77.0)	68.0 (43.0 ÷ 81.0)	57.0 (43,0 ÷ 73,0)	72.5 (43,0 ÷ 78,0)	59.5 (41.0 ÷ 83.0)	0.868
DBP ** pc ^1^	88.0 (81.0 ÷ 96.0)	92.5 (72.5 ÷ 97.0)	83.0 (56.0 ÷ 89.0)	89.5 (67.0 ÷ 96.0)	92.5 (74.0 ÷ 98.0)	0.229
Blood pressure ≥ 95 pc ^1^	8% (2)	19% (10)	9% (2)	12% (3)	17% (3)	0.647

* SBP—systolic blood pressure. ** DBP—diastolic blood pressure. ^1^ pc—percentile. ^2^ NBL—neuroblastoma. ^3^ WT—Wilms tumor.

**Table 4 children-10-01073-t004:** Biochemical parameters in the control group and in the survivor groups; the median results (25Q ÷ 75Q) are provided.

	Control Group	Hematopoietic and Lymphoproliferative Malignancies	NBL ^1^	WT ^2^	Other Solid Tumors	*p*
Salusin β (pg/mL)	327.0 (227.0 ÷ 407.0)	100.0 (61.5 ÷ 181.5)	96.0 (72.9 ÷ 277.0)	227.5 (122.4 ÷ 333.0)	143.0 (110.8 ÷ 271.0)	0.000 *
Total cholesterol (mg/dL)	148.0 (135.0 ÷ 171.0)	152.0 (136.0 ÷ 174.0)	162.0 (149.0 ÷ 168.0)	156.0 (145.0 ÷ 170.0)	166.5 (141.5 ÷ 188.5)	0.394
LDL ^3^ (mg/dL)	86.0 (71.0 ÷ 96.0)	89.0 (74.0 ÷ 106.0)	91.0 (77.0 ÷ 102.0)	87.0 (78.0 ÷ 103.0)	97.0 (79.0 ÷ 117.0)	0.513
HDL ^4^ (mg/dL)	50.0 (43.0 ÷ 61.0)	49.0 (43.0 ÷ 56.0)	54.0 (40.0 ÷ 60.0)	56.0 (48,0 ÷ 58,0)	51.5 (36.0 ÷ 67.0)	0.576
TG ^5^ (mg/dL)	69.0 (48.0 ÷ 89.0)	62.0 (50.0 ÷ 85.0)	83.0 (57.0 ÷ 91.0)	67.0 (55.0 ÷ 83.0)	79.0 (57.0 ÷ 123.0)	0.315
Glucose (mg/dL)	86.0 (82.0 ÷ 89.0)	87.0 (81.0 ÷ 89.0)	82.0 (79.0 ÷ 87.0)	83.0 (81.0 ÷ 86.0)	85.0 (80.0 ÷ 88.0)	0.133
Uric acid (mg/dL)	4.25 (3.40 ÷ 5.00)	5.00 (4.10 ÷ 5.50)	4.00 (3.80 ÷ 4.80)	4.90 (4.55 ÷ 5.40)	3.55 (3.00 ÷ 4.70)	0.001 *
Fibrinogen (g/L)	2.48 (2.30 ÷ 2.85)	2.47 (2.26 ÷ 2.98)	2.57 (2.31 ÷ 2.96)	2.97 (2.53 ÷ 3.48)	2.82 (2.34 ÷ 2.97)	0.473

* Statistically significant differences. ^1^ NBL—neuroblastoma. ^2^ WT—Wilms tumor. ^3^ LDL—low-density lipoprotein. ^4^ HDL—high-density lipoprotein. ^5^ TG—triglycerides.

**Table 5 children-10-01073-t005:** Kidney parameters in the control group and in the survivor groups; the median results (25Q ÷ 75Q) are given.

	Control Group	Hematopoietic and Lymphoproliferative Malignancies	NBL ^4^	WT ^5^	Other Solid Tumors	*p*
Creatinine _s_ (mg/dL)	0.530 (0.450 ÷ 0.730)	0.635 (0.520 ÷ 0.730)	0.555 (0.440 ÷ 0.640)	0.610 (0.540 ÷ 0.715)	0.665 (0.580 ÷ 0.840)	0.040 *
eGFR ^1^ (mL/min/1.73 m^2^)	119.5 (98.2 ÷ 141.4)	108.4 (96.7 ÷ 123.2)	111.5 (98.8 ÷ 122.0)	104.1 (94.8 ÷ 114.3)	93.7 (78.5 ÷ 104.8)	0.009 *
eGFR ^2^ (mL/min/1.73 m^2^)	98.5 (85.2 ÷ 117.0)	91.6 (84.3 ÷ 101.1)	93.4 (83.9 ÷ 99.7)	89.0 (83.7 ÷ 93.9)	79.6 (73.3 ÷ 87.1)	0.004 *
Cystatin C (mg/L)	0.89 (0.74 ÷ 0.99)	0.77 (0.7 ÷ 0.82)	0.79 (0.7 ÷ 0.91)	0.97 (0.85 ÷ 1.13)	0.995 (0.93 ÷ 1.06)	0.000 *
eGFR ^3^ (mL/min/1.73 m^2^)	104.4 (92.7 ÷ 128.5)	122.9 (114.5 ÷ 136.8)	120.3 (101.9 ÷ 136.8)	95.4 (80.3 ÷ 110.0)	92.2 (85.8 ÷ 99.4)	0.000 *
BUN (mg/dL)	11.0 (8.0 ÷ 12.0)	11.5 (9.6 ÷ 13,4)	13.0 (10.0 ÷ 15.0)	12.8 (11.0 ÷ 14.6)	13.3 (11.9 ÷ 16.1)	0.006 *
Albumin in urine (mg/L)	7.4 (5.3 ÷ 10.5)	9.2 (5.0 ÷ 17.4)	12.1 (5.7 ÷ 16.0)	7.0 (5.0 ÷ 10.0)	23.4 (13.5 ÷ 93.0)	0.003 *
Creatinine _u_ (mg/dL)	117.6 (80.9 ÷ 166.2)	101.8 (53.6 ÷ 163.5)	104.3 (63.3 ÷ 130.1)	87.6 (54.7 ÷ 139.0)	85.9 (51.8 ÷ 152.1)	0.710
ACR (mg/g creatinine)	7.9 (4.8 ÷ 13.1)	9.4 (5.9 ÷ 19.2)	14.8 (8.8 ÷ 17.8)	9.2 (7.4 ÷ 15.1)	24.8 (16.6 ÷ 86.7)	0.018 *

* Statistically significant differences. ^1^ eGFR based on the revised Schwartz equation. ^2^ eGFR based on the creatinine and BUN equation. ^3^ eGFR based on the Filler equation. ^4^ NBL—neuroblastoma. ^5^ WT—Wilms tumor. _s_ serum. _u_ urine.

**Table 6 children-10-01073-t006:** Kidney parameters according to ultrasound in the control group and in the survivor groups.

	Control Group	Hematopoietic and Lymphoproliferative Malignancies	NBL ^2^	WT ^3^	Other Solid Tumors	*p*
Left kidney pc ^1^ (age)	<2.5—0% 2.5–10—0% 10–25—8% 25–50—20% 50—0% 50–75—0% 75—4% 75–90—24% 90–97.5—12% >97.5—8%	<2.5—2.7% 2.5–10—0% 10–25—2.7% 25–50—35.1% 50—0% 50–75—18.9% 75—2.7% 75–90—16.2% 90–97.5—5.4% >97.5—16.2%	<2.5—0% 2.5–10—9.1% 10–25—9.1% 25–50—18.2% 50—0% 50–75—18.2% 75—9.1% 75–90—18.2% 90–97.5—9.1% >97.5—9.1%	<2.5—9.1% 2.5–10—0% 10–25—0% 25–50—0% 50—0% 50–75—9.1% 75—0% 75–90—0% 90–97.5—0% >97.5—81.8%	<2.5—28.6% 2.5–10—7.1% 10–25—14.3% 25–50—14.3% 50—7.1% 50–75—0% 75—0% 75–90—14.3% 90–97.5—0% >97.5—14.3%	0.001 *
Left kidney pc ^1^ (height)	<2.5—0% 2.5–10—0% 10—0% 10–25—4% 25–50—24% 50–75—28% 75–90—28% 90–97.5—16% >97.5—0%	<2.5—2.6% 2.5–10—0% 10—0% 10–25—5.3% 25–50—28.9% 50–75—21.1% 75–90—13.2% 90–97.5—15.8% >97.5—13.2%	<2.5—25% 2.5–10—0% 10—0% 10–25—0% 25–50—13.6% 50–75—17.4% 75–90—7.7% 90–97.5—8.3% >97.5—5.9%	<2.5—9.1% 2.5–10—0% 10—0% 10–25—0% 25–50—0% 50–75—18.2% 75–90—0% 90–97.5—0% >97.5—72.7%	<2.5—7.1% 2.5–10—7.1% 10—7.1% 10–25—21.4% 25–50—14.3% 50–75—14.3% 75–90—0% 90–97.5—7.1% >97.5—21.4%	0.001 *
Right kidney pc ^1^ (age)	<2.5—0% 2.5–10—0% 10–25—4% 25–50—28% 50–75—12% 75—4% 75–90—28% 90—4% 90–97.5—12% >97.5—8%	<2.5—2.6% 2.5–10—0% 10–25—2.6% 25–50—18.4% 50–75—28.9% 75—0% 75–90—21.1% 90—2.6% 90–97.5—13.2% >97.5—10.5%	<2.5—10% 2.5–10—0% 10–25—0% 25–50—10% 50–75—20% 75—0% 75–90—10% 90—0% 90–97.5—40% >97.5—10%	<2.5—9.1% 2.5–10—0% 10–25—0% 25–50—9.1% 50–75—0% 75—0% 75–90—9.0% 90—0% 90–97.5—0% >97.5—72.7%	<2.5—14.3% 2.5–10—7.2% 10–25—21.4% 25–50—14.3% 50–75—14.3% 75—0% 75–90—0% 90—0% 90–97.5—0% >97.5—28.6%	0.002 *
Right kidney pc ^1^ (height)	<2.5—0% 2.5–10—4% 10—0% 10–25—12% 25–50—8% 50–75—24% 75–90—40% 90–97.5—4% >97.5—8%	<2.5—2.6% 2.5–10—0% 10—0% 10–25—7.9% 25–50—39.5% 50–75—7.9% 75–90—23.7% 90–97.5—13.2% >97.5—5.3%	<2.5—0% 2.5–10—8.3% 10—16.7% 10–25—0% 25–50—33.3% 50–75—8.3% 75–90—6.7% 90–97.5—16.7% >97.5—0%	<2.5—0% 2.5–10—9.1% 10—% 10–25—0% 25–50—0% 50–75—0% 75–90—9.1% 90–97.5—0% >97.5—81.8%	<2.5—14.3% 2.5–10—21.4% 10—0% 10–25—7.1% 25–50—21.4% 50–75—7.1% 75–90—14.3% 90–97.5—7.1% >97.5—7.1%	0.000 *
RI of the left kidney	0.61 (0.60 ÷ 0.66)	0.65 (0.62 ÷ 0.68)	0.63 (0.60 ÷ 0.65)	0.67 (0.59 ÷ 0.70)	0.64 (0.60 ÷ 0.67)	0.275
RI of the right kidney	0.62 (0.60 ÷ 0.66)	0.64 (0.60 ÷ 0.66)	0.62 (0.61 ÷ 0.65)	0.65 (0.60 ÷ 0.67)	0.64 (0.62 ÷ 0.66)	0.588

* Statistically significant differences. In brackets—median results (25Q ÷ 75Q). ^1^ pc—percentile. ^2^ NBL—neuroblastoma. ^3^ WT—Wilms tumor.

**Table 7 children-10-01073-t007:** Components of metabolic syndrome.

	Metabolic Syndrome	Obesity + HDL ≤ 40 mg/dL	Obesity + Elevated Blood Pressure	Obesity + TG ≥ 150 mg/dL	HDL ≤ 40 md/dL + TG ≥ 150 mg/dL	Elevated Blood Pressure + glc ≥ 100 md/dL	Elevated Blood Pressure + HDL ≤ 40 md/dL
Patients	2	9 WT 1 Hepatoblastoma 1 ALL 2 NBL 2 RMS 1 NHL 1 Medulloblastoma 1	4 ALL 3 NBL 1	1 ALL	1 NBL	1 ALL	0
Control group	0	1	0	0	0	0	1

## Data Availability

Not applicable.

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
