# Peer review of "Assessment of Metabolic Syndrome and Kidney and Heart Function in Childhood Cancer Survivors"

_children, 2023, doi:10.3390/children10061073_

Round 1

Reviewer 1 Report

This study investigated prognostic factors of developing atherosclerosis before developing of symptomatic cardiovascular diseases. The rigor of the study is weakened due to the following issues.

1.       The title of this study did not explicitly reflect the objectives of the study.

2.       Background information is not comprehensive enough to bring about the significance of the problem investigated. Statistics about the issues could be provided to allow the readers to realise the prevalence of those long-time side effects experienced by childhood cancer survivors.

3.       The research gap was not well-justified.

4.       How was the sample size calculated?

5.       The results were not clearly presented.

Reviewer 2 Report

Dear Authors

This study examined to components of metabolic syndrome, kidney and heart function in childhood cancer survivors. The manuscript has been well designed and written. It is very interesting study and it presents good data on this very important topic. I believe that this is very excellent issue in field of medicine section. Moreover, this topic was not clearly revealed in the field of medicine section. So, I would like to thank the authors for their work on this manuscript.

Minor concerns

Abstract

Please delete the (numbers) and then change to bold.

(1) Background -> Background

(2) Methods -> Methods

(3) Results -> Results

(4) Conclusions -> Conclusions

Whole manuscript.

Please, refer to author’s guideline and keep the journal formatting of manuscript.

The reference format has to change from (number) to [number].

Introduction section.

There has a little literature background in Introduction section.

You should add the background of metabolic syndrome, kidney, and heart function in childhood cancer survivors. Namely, your manuscript has only two paragraphs. You should add more 4-6 paragraphs in Introduction section

Line 59

Please change from “hematopoietic malignances (52), neuroblastomas (22), Wilms tumors (24) and other solid tumors (18).” to “hematopoietic malignances (n=52), neuroblastomas (n=22), Wilms tumors (n=24) and other solid tumors (n=18).”

Line 76

Please change from “using standard techniques (Mensor WE 150, 2014).” To “using standard techniques [reference number] or using standard techniques [model name, company name, city, country]”.

In Table 1 and Table 3, “Weight (pc), Height (pc), BMI (pc), <90pc/>90pc etc.”

What does mean “pc”????

In Table 1, change from “p=0.288” to “0.288”

In Table 2 ,3, and 4, you should describe full name of NBL and WT in end of Table 2 and 3.

In Table 4, you should describe full name of LDL, HDL, and TG in end of Table 4.

There are too many abbreviations in all Tables. You have to describe full name of abbreviations in all Tables and main text.

Discussion section - In general, the Introduction section is quite good.

References section - You have to change references’ format in whole manuscript.

I recommend that this manuscript should be edited by an English professional editor for more readable. There are too many typo and grammatical errors.

Round 2

Reviewer 1 Report

Thanks for revising the manuscript.

Reviewer 2 Report

I congratulate the authors for conducting such an interesting study. Although several limitations are still present in the research design based on three reviewers, the authors edited various parts of manuscript as suggested. For this reason, I would like to inform that the article is now appropriate for publication in this journal (Accept).